# OpenReview forum: "Towards Heterogeneous Long-tailed Learning: Benchmarking, Metrics, and Toolbox"
_NeurIPS.cc/2024/Datasets_and_Benchmarks_Track — NeurIPS 2024 Track Datasets and Benchmarks Poster_

### Official Review · Reviewer_xqDp · 2024-06-22
**A worthwhile new benchmark for long tail**

**Rating:** 8
**Confidence:** 4
**Correctness:** Yes
**Clarity:** Yes

**Review:**

In my opinion this is an excellent new tool that will help the research community deal with an issue that is pervasive, but has received inadequate attention.

The paper is well written, the argument for this benchmark well presented and persuasive.  And the construction of the benchmark is sound and fairly comprehensive.

**Strengths:**

This is a problem that matters, yet has received a level of attention substantially less than it pervasive and problematic nature warrants.

The paper is well written, the argument for this benchmark well presented and persuasive.  And the construction of the benchmark is sound and fairly comprehensive.

**Additional Feedback:**

None

**Documentation:**

Yes

**Ethics:**

I do not suspect any ethical concerns.

**Limitations:**

This is an area that could be improved.

**Opportunities For Improvement:**

Could have better addressed limitations and possible future improvements to this benchmark.

**Relation To Prior Work:**

Yes.

**Summary And Contributions:**

The authors present a new benchmark for learning on long-tail data.  They present a methodology for combining metrics to better assess and characterize the long-tailedness of a dataset.  And they propose a benchmark which covers  multiple types of data (tabluar, sequential, grid/image, relational/graph) and multiple problem domains.   They evaluate a set of SOTA algorithms using this benchmark.

---

> ### Author Rebuttal · Authors · 2024-08-17
>
> >**Q.** Could have better addressed limitations and possible future improvements to this benchmark.
>
> **A:** Thank you for your valuable feedback. We appreciate your support and are committed to continually updating and maintaining our GitHub repository to address these concerns. We summarized the constructive comments from all reviews and provided to-do items below:
> - **Expanding benchmark comparisons:** We aim to include additional comparisons across more application domains, such as remote sensing data `[1]` and imbalanced regression `[2]`.
>
> | Dataset  | Method    | MAE  |        |       |         | GMean |        |      |         |
> |----------|-----------|------|--------|-------|---------|-------|--------|------|---------|
> |          |           | Many | Medium | Few   | Overall | Many  | Medium | Few  | Overall |
> |AgeDB`[2]`|SmoteR`[3]`| 7.39 | 8.65   | 12.28 | 8.16    | 4.65  | 5.69   | 8.49 | 5.21    |
> |          |SMOGN`[4]` | 7.64 | 9.01   | 12.09 | 8.26    | 4.90  | 6.19   | 8.44 | 5.36    |
>
> | Dataset    | Method   | Acc  |        |      |         | Precision | Recall | MAP  | Time (s) |
> |------------|----------|------|--------|------|---------|-----------|--------|------|----------|
> |            |          | Many | Medium | Few  | Overall |           |        |      |          |
> |FGSC-23`[1]`|SHIKE`[5]`| 67.5 | 71.4   | 70.0 | 68.5    | 68.2      | 70.0   | 50.8 | 4.55     |
>
> - **Incorporating state-of-the-art algorithms:** We will continue to integrate state-of-the-art long-tailed learning algorithms, such as SHIKE `[5]`.
>
> |Dataset|Method|Acc|Precision|Recall|MAP|Time(s)|
> |-|-|-|-|-|-|-|
> |CIFAR-100-LT(IF=10)|OLTR|12.5|12.0|12.5|3.2|12.9|
> ||BALMS|55.2|54.8|55.2|33.6|2.2|
> ||TDE|58.9|58.9|59.7|36.8|**1.7**|
> ||Decoupling|51.5|51.5|51.4|29.2|1.8|
> ||BBN|59.7|59.9|59.7|38.2|4.3|
> ||MiSLAS|63.3|63.4|63.2|42.3|7.2|
> ||PaCo|66.0|66.2|66.0|46.0|2.5|
> ||SHIKE`[5]`|**68.9**|**69.0**|**68.9**|**68.9**|49.6|
>
> - **Introducing two new metrics:** We will introduce two long-tailedness metrics BI3 `[6]` and CCDF, to provide a more comprehensive understanding of the data distribution. The detailed statistics and results can be found in the **following table and Figure 1 in the attached PDF**.
>
> | Dataset            | IF      | Gini  | Pareto |BI3`[6]`|
> |--------------------|---------|-------|--------|--------|
> | Cora-Full          | 62      | 0.321 | 0.919  | 0.697  |
> | Wiki               | 45      | 0.414 | 1.000  | 0.633  |
> | Amazon-Clothing    | 10      | 0.343 | 0.814  | 0.129  |
> | Amazon-Electronics | 9       | 0.329 | 0.600  | 0.052  |
>
> - **Enabling community contributions:** We plan to allow other users to contribute to the repository.
>
> `[1]` Tang, Haojun, et al. "Text-Guided Diverse Image Synthesis for Long-Tailed Remote Sensing Object Classification." IEEE Transactions on Geoscience and Remote Sensing, 2024.
>
> `[2]` Yang, Yuzhe, et al. "Delving into deep imbalanced regression." International conference on machine learning. PMLR, 2021.
>
> `[3]` Torgo, Luís, et al. "Smote for regression." Portuguese conference on artificial intelligence. Berlin, Heidelberg: Springer Berlin Heidelberg, 2013.
>
> `[4]` Branco, Paula, Luís Torgo, and Rita P. Ribeiro. "SMOGN: a pre-processing approach for imbalanced regression." First international workshop on learning with imbalanced domains: Theory and applications. PMLR, 2017.
>
> `[5]` Jin, Yan, et al. "Long-tailed visual recognition via self-heterogeneous integration with knowledge excavation." Proceedings of the IEEE/CVF conference on computer vision and pattern recognition. 2023.
>
> `[6]` Lu, Yang, Yiu-Ming Cheung, and Yuan Yan Tang. "Bayes imbalance impact index: A measure of class imbalanced data set for classification problem." IEEE transactions on neural networks and learning systems, 2019.

---

### Official Review · Reviewer_UBRj · 2024-07-20
**A good benchmark paper that will draw interests in the community**

**Rating:** 7
**Confidence:** 4
**Correctness:** This paper is technically sound.
**Clarity:** Yes. The paper is well-organized and …

**Review:**

Pros:
1.	There is no existing or similar work addressing the benchmark and toolbox for long-tail learning, although long-tail learning has been extensively studied in the past 3-5 years.
2.	The 6 Datasets across 3 Domains are comprehensive to cover the field of long-tail learning, especially for readers that only deal with one of the areas.
Cons:
1.	The title if called long-tailed learning, but the content is mainly about classification. Actually, there is also work addressing the problem of imbalanced regression. The authors should either discuss imbalanced regression or replace the title by long-tailed classification.
2.	Some types of data that are extensively studied and also suffer from class imbalance are missing, such as remote sensing data and point cloud data.
3.	In Angle 3, all of the tasks are about classification. The first one is just called “classification” without any prefix. It should be denoted by a more specific title.
4.	In Table 4, the algorithms available in HeroLT are a little dated. The latest one was proposed in 2022, but we are in 2024 now. Some recent SOTA work like SHIKE (CVPR 2023) can also be involved:

**Strengths:**

See Pros in Review.

**Additional Feedback:**

N/A

**Documentation:**

The authors provide detailed documentation and Github.

**Limitations:**

This work has no potential negative societal impact.

**Opportunities For Improvement:**

1.	Please continue to update and maintain the GitHub repository as new long-tailed data and tasks become available. Additionally, it’s highly recommended to consider enabling permissions to allow other users to contribute to the repository, fostering a collaborative and dynamic development environment.
2.	For Sec. 2.2, it is better to introduce and discuss the three angles. How can they support the structure of HeroLT? This is crucial for a survey paper.
3.	In angle 1, there are also some work addressing the data complexity factor, e.g. Bayes Imbalance Impact Index: A Measure of Class Imbalanced Dataset for Classification Problem (TNNLS 2020). In addition to the three metrics, the authors should also mention the data complexity factor considering class imbalance.
4.	Text data can be considered to be separated from sequential data, because there are a wide range of NLP tasks suffer from data imbalance.

**Relation To Prior Work:**

Yes. This work covers appropriate related works.

**Summary And Contributions:**

This paper provides a systematic view of long-tailed learning, a recently hot topic mainly in computer vision. This paper summarizes the commonly data and tasks in long-tailed learning with a well-organized structure. So far, there’s no similar work that is as comprehensive as this paper. I believe the benchmark and toolbox will be interested in the community.

---

> ### Author Rebuttal · Authors · 2024-08-17
>
> >**Q1.** Some types of data that are extensively studied and also suffer from class imbalance are missing, such as remote sensing data and point cloud data.
>
> **A1:** Thank you for pointing this out. We will conduct additional experiments on remote sensing data `[1]` and point cloud data `[2]` by comparing state-of-the-art baselines in our benchmark. The preliminary results show the results of the SHIKE algorithm `[3]` on the remote sensing FGSC-23 dataset `[1]`. We are running additional experiments and will update them upon their completion.
>
> |Dataset|Task|Method|Acc||||Precision|Recall|MAP|Time(s)|
> |-|-|-|-|-|-|-|-|-|-|-|
> ||||Many|Medium|Few|Overall|||||
> |FGSC-23`[1]`|Imageclassification|SHIKE|67.5|71.4|70.0|68.5|68.2|70.0|50.8|4.55|
>
> `[1]` Tang, Haojun, et al. "Text-Guided Diverse Image Synthesis for Long-Tailed Remote Sensing Object Classification." IEEE Transactions on Geoscience and Remote Sensing. 2024.
>
> `[2]` Li, Mengke, Yiu-ming Cheung, and Yang Lu. "Long-tailed visual recognition via gaussian clouded logit adjustment." CVPR, 2022.
>
> `[3]` Jin, Yan, et al. "Long-tailed visual recognition via self-heterogeneous integration with knowledge excavation." CVPR, 2023.
>
> >**Q2.** In Table 4, the algorithms available in HeroLT are a little dated. Some recent SOTA work like SHIKE can also be involved.
>
> **A2:** Thank you for pointing this out. We will include discussions on SOTA algorithms `[3,4,5]` and integrate SHIKE `[3]` to our benchmark with additional experiments. Preliminary results show that SHIKE outperforms other algorithms on the CIFAR-100-LT dataset with an imbalance factor of 10, but it takes a longer time. We are running additional experiments and will update them upon their completion.
>
> |Dataset|Method|Acc|Precision|Recall|MAP|Time(s)|
> |-|-|-|-|-|-|-|
> |CIFAR-100-LT(IF=10)|OLTR|12.5|12.0|12.5|3.2|12.9|
> ||BALMS|55.2|54.8|55.2|33.6|2.2|
> ||TDE|58.9|58.9|59.7|36.8|**1.7**|
> ||Decoupling|51.5|51.5|51.4|29.2|1.8|
> ||BBN|59.7|59.9|59.7|38.2|4.3|
> ||MiSLAS|63.3|63.4|63.2|42.3|7.2|
> ||PaCo|66.0|66.2|66.0|46.0|2.5|
> ||SHIKE`[3]`|**68.9**|**69.0**|**68.9**|**68.9**|49.6|
>
> `[4]` Lu, Yang, et al. "Label-noise learning with intrinsically long-tailed data." Proceedings of the IEEE/CVF International Conference on Computer Vision. 2023.
>
> `[5]` Li, Mengke, et al. "Feature Fusion from Head to Tail for Long-Tailed Visual Recognition." Proceedings of the AAAI Conference on Artificial Intelligence. 2024.
>
> >**Q3.** In angle 1, in addition to the three metrics, the authors should also mention the data complexity factor, e.g. Bayes Imbalance Impact Index.
>
> **A3:** Thank you for pointing this out. We would like to clarify that **Pareto-LT** we introduced jointly evaluates data imbalance and the number of categories, and can characterize data complexity to some content.
>
> In addition, we will **integrate Bayes Imbalance Impact Index (BI3 `[6]`) and Complementary Cumulative Distribution Function (CCDF)** raised by the reviewers into our toolbox. The implementation is available on our GitHub under `/HeroLT/tools/evaluations.py`. Detailed statistics and results are provided in **Table 1 and Figure 1 in the attached PDF**. It is worth noting that the BI3 metric is designed for binary classification in the original paper, we adapt the BI3 metric to our long-tailed setting with a large number of categories. Although BI3 is a proper index to describe how much improvement can be made by applying imbalance recovery methods, the correlation does not consistently hold for the complex long-tailed data. Additionally, CCDF provides an intuitive visual representation of the distribution of dataset labels. However, it is not convenient to directly compare the long-tailedness among different datasets.
>
> `[6]` Lu, Yang, Yiu-Ming Cheung, and Yuan Yan Tang. "Bayes imbalance impact index: A measure of class imbalanced data set for classification problem." IEEE transactions on neural networks and learning systems, 2019.
>
> >**Q4.** The title is called long-tailed learning, the authors should either discuss imbalanced regression or replace the title by long-tailed classification.
>
> **A4:** Thank you for your valuable feedback. We would like to clarify that, in addition to the classification task, we also introduced the **instance segmentation** task (lines 175-179) and conducted related experiments (Table 6).
>
> In addition, we will introduce a discussion on **imbalanced regression** to cover a broader context. Below we present the preliminary results using two methods on AgeDB dataset `[7]`: SmoteR `[8]`, an adaptation of the well-known SMOTE technique for regression; and SMOGN `[9]`, which combines SmoteR with Gaussian noise.
>
> |Dataset|Method|MAE||||GMean||||
> |-|-|-|-|-|-|-|-|-|-|
> |||Many|Medium|Few|Overall|Many|Medium|Few|Overall|
> |AgeDB`[7]`|SmoteR`[8]`|7.39|8.65|12.28|8.16|4.65|5.69|8.49|5.21|
> ||SMOGN`[9]`|7.64|9.01|12.09|8.26|4.90|6.19|8.44|5.36|
>
> `[7]` Yang, Yuzhe, et al. "Delving into deep imbalanced regression." International conference on machine learning. PMLR, 2021.
>
> `[8]` Torgo, Luís, et al. "Smote for regression." Portuguese conference on artificial intelligence. Berlin, Heidelberg: Springer Berlin Heidelberg, 2013.
>
> `[9]` Branco, Paula, Luís Torgo, and Rita P. Ribeiro. "SMOGN: a pre-processing approach for imbalanced regression." First international workshop on learning with imbalanced domains: Theory and applications. PMLR, 2017.
>
> >**Q5.** In Angle 3, the first one is just called “classification” without any prefix. It should be denoted by a more specific title.
>
> **A5:** Thank you for your insightful comment. We agree that the term "classification" could be more specific in this context. We will rename it to "object recognition `[10]`" to avoid any confusion.
>
> `[10]` Mani, Inderjeet, and I. Zhang. "kNN approach to unbalanced data distributions: a case study involving information extraction." Proceedings of workshop on learning from imbalanced datasets, ICML. 2003.

---

### Official Review · Reviewer_9zXC · 2024-07-23
**Review of "Towards Heterogeneous Long-tailed Learning: Benchmarking, Metrics, and Toolbox"**

**Rating:** 6
**Confidence:** 2
**Correctness:** Yes.
**Clarity:** Yes.

**Review:**

This paper is well-written and easy to understand. It makes two main points: 1) Most research focuses on data imbalance but often ignores the issue of having many categories in long-tailed distributions. 2) No single algorithm performs best in every task and domain, which shows that choosing the right algorithm for each situation is crucial. This finding is important and should be explored further.

**Strengths:**

This paper provides a fair and accessible performance evaluation of 16 state-of-the-art methods across multiple benchmark datasets using both accuracy-based and ranking-based metrics. This evaluation can be a valuable resource in the field. The paper introduces HEROLT, a comprehensive benchmark for heterogeneous long-tailed learning. HEROLT includes 16 state-of-the-art algorithms and 6 evaluation metrics tested on 16 real-world datasets across 5 tasks and 3 domains.

**Additional Feedback:**

N/A.

**Documentation:**

The availability, maintenance, and ethical use of the dataset are not clearly stated.

**Ethics:**

N/A.

**Limitations:**

Yes.

**Opportunities For Improvement:**

The benchmark and experiments are clearly presented. However, it is still necessary to include a study on data distribution to provide a more comprehensive understanding.

**Relation To Prior Work:**

Yes.

**Summary And Contributions:**

This paper explores three key aspects of long-tailed learning:

1) Characterization of Long-Tailed Data (A1): Long-tailed data is characterized by a highly skewed distribution with many categories, where some categories are significantly underrepresented. 2) Data Complexity Across Domains (A2): Various complex domains, such as tabular, sequential, grid, and relational data, often exhibit long-tailed distributions. 3) Heterogeneity of Emerging Tasks (A3): The paper emphasizes the importance of evaluating the applicability and limitations of existing methods across diverse and heterogeneous tasks.

To address these aspects, we designed extensive experiments involving 16 state-of-the-art algorithms and 6 evaluation metrics, tested on 16 real-world benchmark datasets spanning 5 tasks from 3 different domains.

---

> ### Author Rebuttal · Authors · 2024-08-17
>
> >**Q.** It is still necessary to include a study on data distribution to provide a more comprehensive understanding.
>
> **A:** Thank you for your insightful suggestion.
> - **Metric for data distribution.** In our paper, we introduce a **generic approach to measuring the long-tailedness of data distributions**. Specifically, we introduce Pareto-LT Ratio, a metric that jointly evaluates data imbalance and the number of categories, offering an advantage over existing metrics like Imbalance Factor and Gini Coefficient, which focus primarily on data imbalance and may not reflect the number of categories.
> - **Experimental studies.** In addition to the three metrics, we will consider **introducing two additional metrics (Bayes Imbalance Impact Index `[1]` and Cumulative Distribution Function)** raised by other reviewers to provide a more comprehensive understanding of long-tailedness of the datasets in our benchmark. The detailed statistics and results can be found in the **following table and Figure 1 in the attached PDF**.
>
> |Dataset|IF|Gini|Pareto|BI3`[1]`|
> |-|-|-|-|-|
> |Cora-Full|62|0.321|0.919|0.697
> |Wiki|45|0.414|1.000|0.633|
> |Amazon-Clothing|10|0.343|0.814|0.129|
> |Amazon-Electronics|9|0.329|0.600|0.052|
>
> In summary, all these metrics have been integrated into our toolbox to facilitate the investigation of data distribution and provide a deeper understanding of the long-tailedness of input data. The implementation is available on our GitHub under `/HeroLT/tools/evaluations.py`.
>
> `[1]` Lu, Yang, Yiu-Ming Cheung, and Yuan Yan Tang. "Bayes imbalance impact index: A measure of class imbalanced data set for classification problem." IEEE transactions on neural networks and learning systems, 2019.

---

> > ### Comment · Reviewer_9zXC · 2024-08-25
> > **Review of "Towards Heterogeneous Long-tailed Learning: Benchmarking, Metrics, and Toolbox"**
> >
> > Thank you for the response. I will keep my score unchanged.

---

### Official Review · Reviewer_smqd · 2024-07-28
**This paper gives comprehensive evaluation of existing works, but is not novel enough.**

**Rating:** 5
**Confidence:** 5
**Correctness:** NA
**Clarity:** See above.

**Review:**

- The paper is well-structured and easy to understand.
- The paper is well-written and easy to follow.
- Although a lot of evaluation experiments are conducted, the novelty of this paper seems limited, thus the significance of this work is not so impressive.

**Strengths:**

See above.

**Additional Feedback:**

NA

**Documentation:**

NA

**Limitations:**

- Although a lot of evaluation experiments are conducted, this paper only selects some of the existing baselines and uses existing evaluation metrics to get the results. Although some conclusions are obtained, further improvements are lacking and the novelty is not enough. It is more like an engineering/project work.

- No new datasets are proposed.

**Opportunities For Improvement:**

- As said in the paper: "surprisingly none of the algorithms statistically outperforms others across all tasks and domains". If the paper can propose a better baseline to realize this, it would be a novel point.

- Again, the used evaluation metrics are proposed by existing works, if a new, more comprehensive metric can be proposed, that would be better.

**Relation To Prior Work:**

This paper only selects some of the existing baselines and uses existing evaluation metrics to get the results.

**Summary And Contributions:**

This paper tries to evaluate the long-tailed problem from three aspects: the characterization of data long-tailedness, the data complexity of various domains, and the heterogeneity of emerging tasks. Specifically, this paper has proposed HEROLT, a comprehensive long-tailed learning benchmark integrating 16 state-of-the-art algorithms, 6 evaluation metrics, and 16 real-world datasets across 5 tasks from 3 domains. Extensive evaluation experiments are conducted.

---

> ### Author Rebuttal · Authors · 2024-08-17
>
> >**Q1.** No new datasets and better baseline are proposed.
>
> **A1:** Thank you for your comments. While not targeting proposing a new baseline or dataset, we would like to emphasize that one of our main contributions is **benchmarking on existing datasets** and the **development of a comprehensive toolbox** for a novel problem named heterogeneous long-tailed learning (**Figure 1 in the attached PDF**):
>
> (1) It facilitates benchmarking heterogeneous long-tailed learning, where the data are of different types and thus may have different pre-processing. Our toolbox integrates the datasets and enables uniform evaluation (lines 195-198) using a standardized function as given in our Github (`/HeroLT/utils/__init__.py`).
>
> (2) We actively engaged in adapting the SOTA long-tailed learning algorithms for solving multiple tasks like image classification and instance segmentation.
>
> >**Q2.** The used evaluation metrics are proposed by existing works.
>
> **A2:** Thank you for your valuable feedback. We appreciate the suggestion to propose a new evaluation metric. In our study, we employ well-defined metrics including Acc, Precision, Recall, bAcc, and MAP to measure the performances of long-tailed algorithms `[1,2]`. These metrics are widely recognized and effectively capture the key aspects of model performance.
> - **Metric for data distribution.** While we do not propose a new evaluation metric, we introduce a **generic approach to measuring the long-tailedness of data distributions**. Specifically, we introduce Pareto-LT Ratio, a metric that jointly evaluates data imbalance and the number of categories, offering an advantage over existing metrics like Imbalance Factor and Gini Coefficient, which focus primarily on data imbalance and may not reflect the number of categories.
> - **Experimental studies.** In addition to the three metrics, we will consider **introducing two additional metrics (Bayes Imbalance Impact Index `[3]` and Cumulative Distribution Function)** raised by other reviewers. The detailed statistics and results can be found in the **following table and Figure 2 the in attached PDF**.
>
> | Dataset            | IF      | Gini  | Pareto |BI3`[3]`|
> |--------------------|---------|-------|--------|--------|
> | Cora-Full          | 62      | 0.321 | 0.919  | 0.697  |
> | Wiki               | 45      | 0.414 | 1.000  | 0.633  |
> | Amazon-Clothing    | 10      | 0.343 | 0.814  | 0.129  |
> | Amazon-Electronics | 9       | 0.329 | 0.600  | 0.052  |
>
> In summary, all these metrics have been integrated into our toolbox. The implementation is available on our GitHub under `/HeroLT/tools/evaluations.py`. The toolbox facilitates the investigation of data distribution and provides a deeper understanding of the long-tailedness of input data.
>
> `[1]` Zhang, Yifan, et al. "Deep long-tailed learning: A survey." IEEE Transactions on Pattern Analysis and Machine Intelligence 45.9 (2023): 10795-10816.
>
> `[2]` Fu, Yu, et al. "Long-tailed visual recognition with deep models: A methodological survey and evaluation." Neurocomputing 509 (2022): 290-309.
>
> `[3]` Lu, Yang, Yiu-Ming Cheung, and Yuan Yan Tang. "Bayes imbalance impact index: A measure of class imbalanced data set for classification problem." IEEE transactions on neural networks and learning systems, 2019.

---

> > ### Author Response · Authors · 2024-08-26
> >
> > Dear reviewer,
> >
> > We sincerely appreciate the effort you dedicated to reviewing our paper! We hope that we have addressed your questions.
> >
> > As our paper's current average rating is 6.25 and Reviewer 9zXC has replied, your insights are particularly important to us. With 5 days remaining in the discussion period, we would greatly appreciate any feedback you might have. Please let us know if our response clarifies your concerns, and we are open to any discussions on improving our paper.
> >
> > Thank you once again for your time and constructive feedback!
> >
> > Best regards,
> >
> > Authors

---

> > ### Comment · Reviewer_smqd · 2024-08-29
> > **Thanks for your rebuttal.**
> >
> > Although the authors have provided detailed replies to my concerns. I still think the novelty of the paper is limited. It is more like an engineering work. I still insist on my decision. But I would like to change the score to 5.

---

> > > ### Author Response · Authors · 2024-08-29
> > >
> > > Thank you for taking the time to review our rebuttal! We will continue to advance and refine our work based on the feedback we receive. Thank you once again for your constructive feedback.

---

### Decision · Program_Chairs · 2024-09-26

**Decision:**

Accept (Poster)

**Comment:**

The paper evaluates current algorithms for long-tailed classification, a prevalent challenge in computer vision over recent years, with a newly proposed benchmark called HeroLT. This benchmark introduces two features: 1) a new metric for long-tailedness that accounts not only for class imbalance but also for the number of categories, and 2) coverage of various data modalities and heterogeneous classification tasks.

The reviewers have differing opinions, so I carefully read through the entire article. After careful consideration, I agree with reviewer `smqd` that the novelty of this paper is not significant, and the findings are not impressive. My major concerns are as follows:
- The datasets used to evaluate long-tailedness are all sourced from existing works, and this paper merely combines them. However, as a benchmark study, it's crucial to explain why these particular datasets were chosen. Why select one dataset over another? What about the quality of the data? The authors don't address these points, making it seem as though they arbitrarily selected datasets from different fields and merged them. For instance, why was the Glass dataset included? As shown in Table 4, SMOTE (developed in 2002) appears to nearly solve it perfectly.
- The positioning of this benchmark paper is unclear. The title, 'Towards Heterogeneous Long-Tailed Learning,' suggests the introduction of a new long-tailed learning problem. However, the term 'heterogeneity' refers to the inclusion of different tasks or data modalities within the benchmark, rather than addressing a novel problem. Moreover, no algorithms for 'heterogeneous learning' are presented. For instance, fields such as graph learning, out-of-distribution generalization, and federated learning all have connections to the concept of heterogeneity.
- The hyper-parameter selection is insufficient. Upon reviewing the appendix, I noticed that most hyper-parameters were set to the default values from the original papers, which is not ideal. A proper benchmark should include a well-defined and fair testing protocol for hyper-parameter tuning.
- The findings are not impressive. From the paper, the authors simply describe the results, while there's no in-depth analysis provided.
- Some key related works are missing. A simple search revealed one such work, Invariant Feature Learning for Generalized Long-Tailed Classification, which introduces a benchmark called the 'Generalized Long-Tailed Classification (GLT) Benchmarks.'

Given these reasons, I decided to reject this paper this time.

Note from PC: The SAC championed the paper for acceptance